# Photoinitiator Selection and Concentration in Photopolymer Formulations towards Large-Format Additive Manufacturing

**DOI:** 10.3390/polym14132708

**Published:** 2022-07-01

**Authors:** Alex Stiles, Thomas-Allan Tison, Liam Pruitt, Uday Vaidya

**Affiliations:** 1Bredesen Center for Interdisciplinary Research, University of Tennessee, Middle Drive, Knoxville, TN 37996, USA; astiles8@vols.utk.edu; 2Tickle College of Engineering, University of Tennessee, Middle Drive, Knoxville, TN 37996, USA; ttison@vols.utk.edu; 3Haslam College of Business, University of Tennessee, 1000 Volunteer Boulevard, Knoxville, TN 37996, USA; wpruitt6@utk.edu; 4Manufacturing Sciences Division, Oak Ridge National Laboratory, 2350 Cherahala Blvd, Knoxville, TN 37932, USA; 5Institute for Advanced Composites Manufacturing Innovation, 2360 Cherahala Blvd, Knoxville, TN 37932, USA

**Keywords:** large scale, large format, additive manufacturing, photopolymer, photobleaching, BAPO

## Abstract

Photopolymers are an attractive option for large-format additive manufacturing (LFAM), because they can be formulated from structural thermosets and cure rapidly in ambient conditions under low-energy ultraviolet light-emitting diode (UV LED) lamps. Photopolymer cure is strongly influenced by the depth penetration of UV light, which can be limited in the 2–4 mm layer thicknesses typical of LFAM. Photoinitiator (PI) systems that exhibit photobleaching have proven useful in thick-section cure applications, because they generate a photoinitiation wavefront, but this effect is time-dependent. This study investigates the light transmission and through-thickness cure behavior in (meth)acrylate photopolymer formulations with the photobleaching initiator bis(2,4,6-trimethylbenzoyl)-phenylphosphine oxide (BAPO). Utilizing an optical model developed by Kenning et al., lower concentrations (0.1 wt% to 0.5 wt%) of BAPO were predicted to yield rapid onset of photoinitiation. In situ cure measurements under continuous UV LED irradiation of 380 mW/cm^2^ showed that a 0.1 wt% concentration of BAPO achieved peak polymerization rate within 2.5 s at a 3-mm depth. With only 1 s of irradiation at 1.7 W/cm^2^ intensity, the 0.1 wt% BAPO formulation also achieved the highest level of cure of the formulas tested. For an irradiation dose of 5.5 J/cm^2^ at a duration of 3.7 s, cured polymer specimens achieved a flexural strength of 108 MPa and a flexural modulus of 3.1 GPa. This study demonstrates the utility of optical modeling as a potential screening tool for new photopolymer formulations, primarily in identifying an upper limit to PI concentration for the desired cure depth. The results also show that photobleaching provides only a limited benefit for LFAM applications with short (1.0 s to 3.7 s) UV irradiation times and indicate that excess PI concentration can inhibit light transmission even under extended irradiation times up to 60 s.

## 1. Introduction

Large-format additive manufacturing (LFAM) is a nascent technology, emerging as an efficient method for producing structural components and tooling for low-volume-composites manufacturing [1,2,3,4,5,6]. LFAM refers to polymer-extrusion-deposition methods that achieve both large-scale and high-material throughput during printing. Most large format three-dimensional (3D) printers use thermoplastic feedstock, and as of 2016, the Big Area Additive Manufacturing (BAAM) system had a reported maximum deposition rate of 36 kg/h [7,8]. Thermosets such as vinyl ester and polyester have also been used in large format printers such as the Reactive Additive Manufacturing (RAM) system developed at the Oak Ridge National Laboratory’s Manufacturing Demonstration Facility (ORNL-MDF). Some challenges of thermoset polymer printers include: (a) balancing cure time and exotherm temperatures to retain wall integrity with uncured resins, (b) limited build height due to wall collapse, and (c) worker safety hazards from volatiles such as styrene [9,10,11,12].

Photopolymers are a subset of thermosets that cure in seconds vs. minutes or hours [13,14]. In additive manufacturing (AM) methods such as direct ink writing (DIW) [15,16,17,18,19], ultraviolet (UV) lamps can be utilized to cure a photopolymer ink immediately after deposition [20,21,22] (see Figure 1). Unlike thermoset printers, this photopolymer DIW (UV-DIW) prevents wall collapse issues via in situ curing and has been used to print self-supporting spans of at least 8 mm [23]. While most photopolymer and thermoset research in AM is focused on layer heights of 25–500 μm, a typical layer height for the BAAM is 2–4 mm [8]. Stiles et al. formulated vinyl ester photopolymers towards LFAM scale, focusing on monomer selection as an approach to minimize cure gradients. They reported that 3 mm layers could be fully cured with an irradiation time of 5 s under high-intensity UV LED lamps [24].

In photopolymers, a curing reaction generally commences with photoinitiation, when light is absorbed by a photoinitiating system that reacts to form initiating species such as free radicals, cations, acids, bases, etc. [25,26,27,28,29]. The present study investigates free-radical systems, which typically use a photoinitiator (PI) to generate free radicals via either a scission process (Type I) [29,30,31,32] or a hydrogen-abstraction process (Type II) [27,29,31,33,34,35]. For scission-type photoinitiators (PIs) such as BAPO and BDMB, the PI molecule separates into two or more free radicals, and these react with C=C bonds in the monomer to form a reactive monomer radical. The monomer radical then initiates a chain reaction with the monomer in the formulation until a termination reaction causes a polymer chain to stop growing. Chain propagation can also lead to crosslinking when multifunctional monomers are used [27]. Early in a reaction, the predominant termination mechanism is bimolecular termination. As the reaction progresses, the viscosity of the solution increases, and radicals can become occluded in the growing polymer network. This radical trapping is a prevalent termination step for glassy photopolymers such as Bis-EMA, which can become vitrified before all monomer is consumed [36,37,38,39,40].

Since light absorption initiates photopolymer reactions, the reaction commences at the irradiated surface before progressing through the polymer bulk. The depth of cure in a photopolymer system is guided by the Beer–Lambert Law (Equation (1)):(1)A=εbC 
where absorbance A is the product of molar absorptivity ε, the length of the light path b, and the concentration of the absorbing material C. The polymer and PI both absorb light to varying extents depending on the material and wavelength of light. With greater light absorbance, light penetration is restricted, and the depth of cure can be inhibited [41]. This can present a challenge for applications such as LFAM, where it is desirable to maximize a printer’s material throughput rate by minimizing cure time at 2–4 mm typical layer heights.

To overcome the limited light penetration of some photopolymer formulations and improve through thickness cure response, several strategies have been developed for curing thick systems [28,34,40,42]. These strategies include: (1) low optical densities, (2) photobleaching, (3) near-infrared (NIR) upconversion, (4) shadow cure chemistries, and (5) RAFT polymerization.

1Low optical density is a strategy which intentionally reduces the absorbance of the photoinitiating system to allow greater light penetration [28]. Per the Beer–Lambert Law, this can be achieved by reducing PI concentration, selecting a PI with low molar absorptivity in the emission region of the light source, or a combination of the two. This approach was successfully employed to cure a methacrylate system up to 31 cm in depth under a 1.1 W/cm^2^ UV LED light source at 405 nm output [29]. For low optical density to be effective, it is necessary to achieve an optimal PI concentration that allows greater light penetration without being too low for effective curing. This will vary depending on factors such as PI, light source, monomer, fillers, and desired depth of cure [43].2Photobleaching is another common method for improving depth of cure in photopolymers [13,27,28,32,44,45,46]. Conventional, non-photobleaching Type I PIs continue absorbing light after reacting, restricting light penetration throughout the curing reaction [27,41]. By contrast, photobleaching PIs have UV-transparent reaction byproducts, allow deeper light penetration into the polymer as the PI reacts. In a process known as photofrontal polymerization, the dynamic light penetration under continuous UV irradiation spurs a free-radical-initiation wavefront [47,48,49,50] that has been shown to allow cure depths of 52 mm [43]. Photopolymers for dentistry regularly incorporate photobleaching PIs to achieve a depth of cure as high as 6 mm in filled systems [32], and novel photobleaching PIs are actively being developed with improved reactivity in the visible/near UV light range typically utilized by dentists [51,52,53]. Since their reaction byproducts are optically transparent, photobleaching PIs are also advantageous for aesthetically critical applications such as gel coats and photocuring inks [27]. Some photobleaching PIs such as BAPO are particularly efficient in the UVA region, making them useful in applications such as stereolithography (SLA) and digital light projection (DLP) [54,55,56,57] that do not require significant depth of cure.3Upconversion particles are specialized nanoparticles that absorb light at a higher wavelength (such as NIR) and re-emit light at wavelengths that can then be absorbed by PIs [28]. By allowing deeper light penetration at wavelengths that are not initially absorbed by the PIs, cure thickness as high as 137 mm [58] has been demonstrated using this technique. However, such systems use rare-earth metals and are not currently in widespread use [28].4Various photopolymer chemistries have been developed with the ability to cure regions that are not directly exposed to the irradiated light [34,42,59,60], a phenomenon also known as “shadow cure” or “dark cure”. The most common of these are cationic-curing systems, in which UV energy generates a protonic acid to initiate a ring opening of epoxy resins [27]. Cationic systems have longer-lived reactive species than conventional free-radical systems, and methods such as transferable shadow curing demonstrate that these long-lived active centers can be dispersed throughout a polymer to cure specimens up to 20 mm thick [61]. In general, cationic-curing chemistries have slower reactions than free-radical systems, but hybrid cationic/free-radical chemistries can reduce this reaction time [60]. In these hybrid systems, the cationic- and free-radical-curing components (e.g., epoxy and acrylate) should be compatible and cure at similar rates to avoid phase separation [40].Thermal initiators and other latent species have also been incorporated into photopolymers, taking advantage of the exothermic polymerization reaction to spur a photoinduced thermal reaction [28]. Unlike photofrontal polymerization, a thermal frontal polymerization reaction can be self-sustaining through the bulk of a polymer. Lecompere et al. have demonstrated this effect with a hybrid cationic/thermal system in curing up to 20 mm of opaque carbon fiber composite [62,63]. Light can be used as a method of directly heating the polymer surface to trigger a frontal polymerization reaction [64], and direct heating without any PI was recently demonstrated in 3D printing applications [65]. While these technologies show promise for curing thick-polymer systems, a common drawback is that the use of thermal initiators can reduce the storage life of a formula or necessitate process changes such as in-line mixing and/or heating [28].

5More recently, “living” or controlled radical polymerization systems have been a topic of intense research, most notably the reversible addition-fragmentation chain transfer (RAFT) process [66]. The RAFT process generally involves the addition of a thiocarbonylthio compound to a free-radical system (e.g., methacrylates), which controls the reaction such that the all of the polymer chains grow at the same rate [66]. This approach has been shown to yield a narrower molecular weight distribution and glass transition region while also improving monomer conversion [66,67]. The RAFT approach shows promise for a wide variety of applications in photocuring [68] including visible-light curing in 3D printing applications [69] and in NIR irradiation [70]. Commercial formulations are becoming available in dentistry that demonstrably achieve a 4-mm depth of cure with only 3 s of irradiation [67]. While RAFT modification may be possible with Bis-EMA photopolymers, the technology has not yet been employed at the scales required for LFAM applications.

The objective of the present study was to improve upon the Bis-EMA/PETIA photopolymer formulations previously developed by Stiles et al. [24] by adjusting PI type and concentration to improve the through-thickness cure response under simulated LFAM conditions. Of the strategies discussed, upconversion particles, thermal initiators and RAFT were not utilized for the reasons previously mentioned. Likewise, hybrid cationic-curing chemistries would necessitate the addition of both a cationic initiator and epoxy, which could alter the cured polymer’s properties to the point of preventing direct comparison against the previous BDMB formulations [24]. Therefore, the most straightforward options for potentially improving through-thickness cure response were to optimize the PI concentration for low optical density and employ a photoinitiating PI. For LFAM applications, the combination of low formulation cost, ready availability, and improved cure depth makes photobleaching PIs a compelling option when printing large components such as tooling for the composites industry.

The present study focused on BAPO as a photobleaching PI to improve the cure response of Bis-EMA in 3-mm-thick layers. Formulations with PI concentrations of 0.1 wt%, 0.5 wt%, and 1.0 wt% were cured under a high-intensity UV LED lamp with a short irradiation time (<5.0 s), simulating the conditions expected of a large-format UV-DIW 3D printer. For a constant UV dose, the authors hypothesized that the level of cure at a 3-mm depth would improve at BAPO concentrations of 0.5 wt% or lower. For the same PI concentration, BAPO formulations were also predicted to achieve a higher degree of cure at a 3-mm depth than non-photobleaching BDMB control formulations.

## 2. Literature Review

For material extrusion printers such as LFAM systems, Chesser et al. calculated that the material-deposition rate drops with the square of the increase in resolution [8]. There are practical limits to print speed for a given printer, such as the mechanical limitations of a gantry and the momentum effects that decrease resolution in the corners. Since machine speed is limited, LFAM printers maximize throughput by increasing layer height. In formulating photopolymers for LFAM, a critical limiting factor is the exposure time required for full through-cure at layer heights of 2–4 mm [24]. Therefore, deposition rates at these layer heights can be improved by decreasing the UV irradiation time required to achieve full through-cure.

In general, for free-radical-initiated (meth)acrylate photopolymers, increased UV lamp intensity and curing temperature have been demonstrated to increase the overall DC for a given formulation, while also reducing reaction time [71,72,73]. Decker found that increased lamp intensity improved acrylate DC even in ambient conditions. An increase in lamp intensity caused a corresponding rise in exotherm temperature within the polymer, increasing reaction rates [72]. The improved DC associated with increased lamp intensity is often attributed to an excess free volume due to rapid polymerization under high light intensity [74,75]. Unsurprisingly, based on these findings, a standard method for reducing cure time is by increasing lamp intensity, which can range widely from 0.5 mW/cm^2^ to 25 W/cm^2^ or more [71,72,73,76,77,78]. Various filled dental resins can achieve complete cure through 2-mm to 6-mm thicknesses, with irradiation times ranging from 20 s to 40 s and lamp intensities of 400 mW/cm^2^ to 1200 mW/cm^2^ [79,80,81,82]. DiPietro reported a full through-cure in 8-mm-thick fiberglass laminates after 15 s using a photobleaching PI and a high intensity 25 W/cm^2^ spotlight [78]. For LFAM applications, an exposure time of 5 s under a 1.7 W/cm^2^ incident-light intensity has been correlated with a linear deposition rate of 5 mm/s [24].

In selecting PIs, the peak light-absorbing wavelengths of the PI are often matched to the peak wavelengths for the lamp output to maximize photoinitiation rates (see Equation (1)). The two primary options for UV lamp systems are mercury vapor lamps and UV LED lamps. The latter are ideal for LFAM because they are more compact and can cycle rapidly without a ‘warm-up’ period to achieve peak output intensity [60,83,84]. UV LED lamps are predominantly available in the UVA emission range (320 to 400 nm). Phosphine oxides such as TPO, TPO-L, and BAPO all absorb light in the UVA range and are photobleaching, making them strong candidates for thick-polymer systems [27,28,32]. BAPO, in particular, is less sensitive to styrene and is commonly used in photopolymer formulations for curing fiberglass-reinforced composites [13,44,45,46].

While selecting a UVA-lamp system improves light-penetration depth, the PI type and PI concentration also strongly influence light penetration and cure kinetics [27]. High PI concentration can block light from penetrating deeper into a polymer, creating a cure gradient [38,48,49]. The ideal PI concentration varies for a given formulation and desired depth of cure. Lee et al. reported cured gel thickness increased up to 1.2 mm, as PI concentration reduced from 0.25 wt% to 0.02 wt% and UV dose increased from 0.9 J/cm^2^ to 22.3 J/cm^2^ [85]. Lecamp et al. found that a dimethacrylate oligomer with Darocur 1173 cured in thin films exhibited an increased conversion rate with increasing PI concentration up to 1.25 wt% but achieved peak ultimate conversion at 0.75 wt% [86]. However, Lecamp et al. also reported that as PI concentration was reduced below 0.75 wt%, the ultimate degree of conversion and resulting glass-transition temperature (T_g_) dropped. PI concentration must, therefore, achieve a balance in a thick-polymer system. Decreasing PI concentration can improve the depth of cure but can also lead to incomplete curing [43] and poor surface cure [27].

An optical model utilizes the Beer–Lambert Law to predict the light intensity and free radical initiation rates within a photopolymer system. Several optical models have been proposed by Ivanov and Decker [87], Miller et al. [47], and Kenning et al. [48,49,50]. These models are of interest because they do not require direct formulation testing and may, therefore, be useful as a potential formulation pre-screening tool in predicting the effect of PI selection and concentration for a given light input and desired cure depth. The Kenning optical model has been selected for the present study because of its suitability for polychromatic light sources [48,49]. The Kenning model builds on Ivanov and Decker’s initially proposed model based on the Beer–Lambert Law, which predicted the optical density (i.e., absorbance) at a given depth in the polymer, based on PI concentration and PI molar absorptivity. This model was later expanded by Miller et al. [47] to account for light attenuation due to the polymer and photolysis byproducts. The Ivanov/Decker and Miller models assumed monochromatic light sources, but Kenning expanded the Miller model to include polychromatic light sources. As with the previous models, the Kenning model predicts light intensity throughout a polymer at a given irradiation time, and by extension, it also predicts rates of free-radical generation [50]. Since free-radical generation initiates the curing reaction, the Kenning model may provide an indirect means of pre-screening a series of formulations for potential cure response under irradiation from the polychromatic UV LED system in the present study. To the authors’ knowledge, no prior studies have attempted to use the Kenning optical model as a screening tool for formulations in LFAM or other high-intensity (>100 mW/cm^2^), short irradiation (>5 s) formulations for curing thick-polymer systems.

While the Kenning optical model predicts free radical generation, it is limited because it does not include reaction rate coefficients or temperature dependence. These factors are included in cure-kinetics models, which do not predict free-radical generation but instead describe the rate at which polymerization occurs. Such models require experimental input on the extent of curing in the polymer obtained via techniques such as UV differential-scanning calorimetry (Photo-DSC), Fourier transform infrared spectroscopy (FTIR), or dielectric analysis [50,76,88,89]. Of these techniques, FTIR is the most common for monitoring relative C=C bond conversion, due to its capability to capture rapid reactions in real-time [67,90,91,92,93]. Cure-kinetics models are capable of describing termination mechanisms of a reaction and change as the degree of conversion increases [26,89,94]. While cure-kinetic models include light intensity as a factor [72,86,89,95,96], most studies assume thin-film curing and do not account for light attenuation in thick-polymer systems. Saenz-Dominguez and others [50,76,77,88,97] have proposed an empirical autocatalytic cure-kinetics model, which has been demonstrated for photopolymer systems up to 2 mm thick. Such models can provide an accurate description of the state of cure through the depth of a polymer system, but they also require extensive testing of the target formulation. Therefore, when considering a model for use as a pre-screening tool, an optical model such as the Kenning model may be ideal for a thick-polymer system.

The present study investigated the effect of BAPO in formulations for LFAM, assuming 2–4 mm layer heights and high intensity (>100 mW/cm^2^) UV LED light sources under short (<5 s) irradiation times. Formulations were prepared at PI concentrations from 0.1 wt% to 1.0 wt% utilizing BAPO, with BDMB as a control PI, and irradiated at a 3-mm thickness under simulated LFAM conditions. The Kenning optical model was used to predict radical initiation rates, and this was compared against experimental results using FTIR. The primary objectives of this study were (1) to assess the utility of optical modeling as a screening tool in developing formulations for LFAM and (2) to investigate the potential benefit of photobleaching PI systems in LFAM photopolymer formulations.

## 3. Materials and Methods

### 3.1. Light Source and Materials

The lamp was an Altair 75 high-intensity UV LED area lamp (Heraeus Noblelight America LLC, Gaithersburg, MD, USA) with an emitting window of 80 mm × 13 mm and peak irradiance of 3.7 W/cm^2^ at 395 nm. The primary oligomer in this study was bisphenol A ethoxylate dimethacrylate (Bis-EMA, Polynt Composites USA Inc., Carpentersville, IL, USA, a di-functional methacrylate that is synthesized with ethoxylated bisphenol A and methacrylic acid (see Figure 2). The reactive diluent was PETIA, an acrylate with a 50:50 molar mixture of tri- and tetra-acrylate esters of pentaerythritol (Allnex, Frankfurt, Germany). The primary PI was BAPO, supplied as Omnirad 819 (IGM Resins USA Inc., Charlotte, NC, USA), and the control PI was BDMB, a highly reactive alkylaminoacetophenone (Lambson Limited, Wetherby, UK). Figure 2 shows the chemical structures for the materials used in this study, and Appendix A shows the reaction byproducts from the BAPO photoinitiation alpha cleavage.

In the present study, 3-mm-thick photopolymer formulations used BAPO as the primary PI and BDMB as a control PI. The experiments and model can be separated into two parts:(a)Optical Model—PI absorptivity values from UV-Vis spectroscopy results were used to build an optical model and predict photoinitiation rates at a 3-mm depth for BAPO concentrations of 0.1 wt%, 0.5 wt%, and 1.0 wt%. Formulations were then prepared at the same concentrations for both BAPO and BDMB, and 3-mm-thick specimens were irradiated continuously for 60 s while monitoring cure via FTIR. The FTIR results were used to determine the cure rate at a 3-mm depth and compared against the optical model.(b)Pulse Irradiation Response—To simulate the curing conditions of LFAM processing, formulations were also irradiated at short irradiation times of 1.0 s and 3.7 s, and FTIR was used to determine the cure response at a 3-mm depth. The four formulations with the highest degree of C=C conversion under 3.7 s irradiation were selected and further tested for mechanical properties under the same irradiation conditions.

The results of the short irradiation testing were then compared against the optical model to determine the potential value of light absorption as a screening mechanism for LFAM photopolymer formulations.

### 3.2. UV Absorption

Absorption properties of BAPO and BDMB were determined by ultraviolet-visible (UV-Vis) spectroscopy on an Evolution 300 spectrometer (Thermo Fisher Scientific, Waltham, MA, USA). PI was dissolved in methanol at 0.01 wt%, and the solution was tested in a 1400 microliter quartz cuvette with a lid (Thorlabs Inc., Newton, NJ, USA). The scan range was 200 to 600 nm, with a bandwidth of 2.0 nm. Specimens were tested before and after 60 s of UV irradiation. Absorption properties were also determined for the uncured resin to determine $A_{mj}$, with resins dissolved at 5 wt% in spectroscopy-grade acetone (Uvasol^®^, MilliporeSigma, Burlington, MA, USA). These solutions were tested in a 3500-microliter quartz cuvette with a lid from Thorlabs, Inc. The scan range was 300 to 500 nm, with a bandwidth of 2.0 nm. Due to the absorbance range of acetone, only the absorbance properties >325 nm could be accurately determined. $A_{mj}$ values are shown in Appendix A.

### 3.3. Optical Model

For the present work, the optical model developed by Kenning et al. was used to predict the temporal and spatial evolution of light intensity through a 3-mm-thick photopolymer [48,49].

The Kenning model has several experimental inputs specific to the lamp setup and photopolymer formulation. Inputs include incident-light intensity and lamp spectral output, PI molar concentration, the molar-light absorptivity of the PI, and absorptivity of the polymer. In the present study, the spectral output was centered at 398 nm for an Altair 75 UV LED lamp. BAPO was used as the PI with molar concentrations of 0.00267 mol/L, 0.01329 mol/L, and 0.02645 mol/L. These concentrations are equivalent to 0.1 wt%, 0.5 wt%, and 1.0 wt%, respectively. UV-Vis spectroscopy was used to obtain the molar absorptivity for BAPO and Bis-EMA. Once these data were obtained, it was then possible to predict the light intensity and photoinitiation rate for Bis-EMA formulations with BAPO PI before actual formulation work.

The governing equations of the Kenning optical model are shown in Equations (2)–(4):(2)$$\frac{\partial C_{i}\left(z,t\right)}{\partial t}=-\frac{C_{i}\left(z,t\right)}{N_{A}h}\underset{j}{\sum }\left(\frac{\varepsilon_{ij}\phi_{j}I_{j}\left(z,t\right)}{\upsilon_{j}}\right)\;$$
(3)$$\frac{\partial C_{p}\left(z,t\right)}{\partial t}=-\frac{C_{i}\left(z,t\right)}{N_{A}h}\underset{j}{\sum }\left(\frac{\varepsilon_{ij}\phi_{j}I_{j}\left(z,t\right)}{\upsilon_{j}}\right)\;$$
(4)$$\frac{\partial I_{j}\left(z,t\right)}{\partial z}=-\left[\varepsilon_{ij}C_{i}\left(z,t\right)+A_{mj}+\varepsilon_{pj}C_{p}\left(z,t\right)\right]I_{j}$$
where j is an index with a different value for each wavelength of light considered, $C_{i}\left(z,t\right)$ is initiator molar concentration at depth z and time t, a $C_{p}\left(z,t\right)$ is photolysis product molar concentration, and $I_{j}\left(z,t\right)$ is incident-light intensity of a specific wavelength. $\varepsilon_{ij}$ is the initiator Napierian molar absorptivity of a specific wavelength, and $\phi_{j}$ is quantum yield of the PI at a specific wavelength (assumed as 0.2) [47,49]. $N_{A}$ is Avogadro’s number and h is Planck’s constant, while $\upsilon_{j}$ is the frequency of light in units of inverse seconds. $A_{mj}$ is the absorption coefficient of the monomer and the polymer-repeat units.

Miller and Kenning assume that the initial initiator concentration is uniform through the polymer depth at time 0 (see Equation (5)):(5)$$C_{i}\left(z,0\right)=C_{0}\;$$

At any time, the incident intensity is constant on the polymer surface (see Equation (6)):(6)$$I\left(0,t\right)=I_{0}$$

Once Equations (2) and (4) are solved numerically, the resulting concentration and light intensity values can be used to determine the rate of photoinitiation, $R_{i}\left(z,t\right)$ in Equation (7):(7)$$R_{i}\left(z,t\right)=2C_{i}\left(z,t\right)\underset{j}{\sum }{\left[I\left(z,t\right)\right]}_{j}\phi_{j}\varepsilon_{ij}\;$$

$C_{i}\left(z,t\right),\;I_{j}\left(z,t\right),\;\;\mathrm{and}\;R_{i}\left(z,t\right)$ were solved numerically by discretizing and using the method of finite differences in MATLAB software (MathWorks, Natick, MA, USA). In addition to neglecting reaction byproducts, the monomer was assumed to be pure Bis-EMA. PETIA was neglected, as were contaminants such as residual methacrylic acid content. As Miller noted, this model also neglects any change in the molar absorptivity of the monomer during curing [47].

### 3.4. Specimen Preparation

A base-polymer formulation was prepared with 80 parts Bis-EMA to 20 parts PETIA by weight. Formulations were then prepared with BAPO or BDMB at PI concentrations of 0.1, 0.5, and 1.0 wt%. These were mixed under low heat using a hot-plate stirrer for 10 min, then allowed to rest for at least 24 h before UV irradiation at ambient temperatures. All test specimens were cured in a polytetrafluoroethylene (PTFE)-molding-fixture measuring 25 mm in diameter with a mold cavity of 6 mm in diameter and 3 mm in thickness. Each specimen was cured with the mold positioned to center specimens in the emitting window. The specimens were irradiated at 10 mm from the top surface of the photopolymer to the UV-lamp-emitting window.

### 3.5. Continuous Irradiation Testing

The formulations were irradiated for 60 s at an incident intensity of 380 mW/cm^2^ on the polymer surface, as measured using an ILT800 radiometer from International Light Technologies. Radiometry testing was performed to measure transmitted-light intensity, and FTIR was used to measure the degree of cure at a 3-mm depth in the polymer specimens. In the radiometry study, all specimens were tested with a PET film and 1-mm-thick quartz discs covering the top and bottom of the mold to mitigate oxygen inhibition and control sample thickness.

For the FTIR study, testing was conducted on a Nicolet iS50 FTIR Spectrometer with an attenuated total reflectance (ATR) module from Thermo Fisher Scientific. To ensure intimate contact of the cured polymer with the diamond crystal of the ATR, the mold and lamp were positioned directly over the crystal. Specimens were irradiated in situ for 60 s. During irradiation, measurements were taken in real-time (RT-FTIR) in a series of individual scans at 4 cm^−1^ resolution and a gain of 8.0, with a duration of 1.3 s per scan. Using the method previously described by Stiles et al. [24], the degree of conversion (DC) of C=C was calculated as the change in the absorbance ratio of the 1637 cm^−1^ methacrylate/acrylate pendant group peak with 1608 cm^−1^ as an internal reference peak [98]. The DC for Bis-EMA formulations was calculated using Equation (8):(8)$$\%DC=\left[1-\frac{\left(\frac{1637{\;\mathrm{cm}}^{-1}}{1608{\;\mathrm{cm}}^{-1}}\right)\;Peak\;height\;cured}{\left(\frac{1637{\;\mathrm{cm}}^{-1}}{1608{\;\mathrm{cm}}^{-1}}\right)\;Peak\;height\;uncured}\right]\times 100\;$$

As the pendant group C=C bonds react, the absorbance-peak height decreases and the DC increases (see Figure 3).

For comparison against pulse-irradiation testing, an additional FTIR measurement was taken at 10 min post-irradiation for each specimen. This measurement consisted of 64 scans at 4 cm^−1^ resolution, which were averaged into a single spectrum. A minimum of three specimens of each formula were tested for both radiometry and RT-FTIR, with the specimen setup shown in Figure 4.

### 3.6. Pulse Irradiation Testing

Each BAPO and BDMB formulation was irradiated for 1.0 s and 3.7 s, at an incident intensity of 1600 mW/cm^2^ and a total equivalent dose of 1.4 J/cm^2^ and 5.5 J/cm^2^, respectively. Samples were prepared in the same setup as RT-FTIR, irradiated in situ on a diamond ATR accessory. Scans were taken before testing for baseline polymer-peak heights and 10 min post-irradiation, with 64 scans at 4 cm^−1^ resolution.

The continuous and pulse irradiation testing served to identify different aspects of the photopolymer behavior. A lower intensity, 380 mW/cm^2^, was selected in the 60 s irradiation testing to enable sufficient data capture during real-time FTIR to experimentally determine the change in R_p_ over time. This allowed for more direct comparison between the optical model and experimental results. For the pulse-irradiation testing, the highest possible lamp intensity, 1600 mW/cm^2^, was selected to imitate the conditions most desirable for LFAM applications. By maximizing intensity, it was possible to reduce through-cure time with minimum UV-exposure time.

### 3.7. Flexural Testing

Two BAPO formulations and two BDMB formulations were selected for further mechanical testing based on FTIR results. Flexural specimens were prepared with nominal dimensions: 12.7 mm width × 3 mm depth × 58 mm length. Specimens were cured in white acrylic molds at 3.7 s pulse irradiation at 1600 mW/cm^2^ incident-light intensity, with PET film and 1-mm-thick glass-microscope slides on top and bottom. Three-point bend testing was conducted per ASTM D790, with six specimens tested per formula and a 16:1 span-to-depth ratio.

Test data were used to calculate flexural strength, flexural strain, and flexural modulus. Flexural strength was determined as the maximum flexural stress ($\sigma_{fM}$) value. Flexural stress $\left(\sigma_{f}\right)$ was calculated using Equation (9):(9)$$\sigma_{f}=\frac{3PL}{2bd^{2}}\;$$
where P is load, L is support span, b is specimen width, and d is the specimen depth. Flexural strain $\left(\varepsilon_{f}\right)$ was reported as the strain at breaking, which coincided with the peak flexural strain due to the brittle failure of the polymers tested. Flexural-strain values were calculated using Equation (10):(10)$$\varepsilon_{f}=\frac{6Dd}{L^{2}}$$
where D is the maximum outer deflection in the center of the beam, flexural modulus was determined as the chord modulus from the initial straight-line portion of the flexural stress-flexural strain curve, as shown in Equation (11):(11)$$E_{f}=\frac{\sigma_{f2}-\sigma_{f1}}{\varepsilon_{f2}-\varepsilon_{f1}}$$
where the chord modulus was the ratio of the difference in flexural stress and flexural strain taken from points at 1.5% and 0.1% flexural strain. These values were chosen based on preliminary photopolymer testing and consistently fell within the initial linear-elastic region for cured Bis-EMA photopolymers.

### 3.8. Statistical Analysis

For statistical analysis, the multiple comparison of means for one-way ANOVA method was used to determine whether datasets with similar means differed significantly from one another. Tukey’s Honestly Significant Difference Procedure (Tukey HSD) was applied to the datasets shown in Figure 9 and Table 1 using MATLAB software [24].

## 4. Results

### 4.1. Absorption Spectra

As shown in Figure 5, both BAPO and BDMB have some absorptivity in the UV-A range, overlapping the spectral output of the Altair 75 lamp. BDMB is considered a highly reactive PI because of its high molar absorptivity at lower wavelengths. However, in the UV-A range, BAPO has higher absorptivity than BDMB. At the peak lamp-output wavelength of 398 nm, BDMB has a molar absorptivity of 249.9 L mol^−1^ cm^−1^ while BAPO is 679.9 L mol^−1^ cm^−1^ (see Figure 5). This higher absorptivity blocks light transmission and can impede the depth of cure during initial irradiation. As indicated by the “Post UV” values in Figure 5, BAPO has UV transparent photoproducts, which allows deeper light penetration into a specimen as the PI reacts. BDMB, on the other hand, has limited photobleaching and nearly no change in absorptivity within the spectral output of the lamp in the present study.

### 4.2. Optical Model for Photobleaching

Based on UV-Vis results, the “Post UV” molar absorptivity shown in Figure 5 indicates that BAPO has reaction byproducts that are UV transparent in the 350 to 450 nm range. Therefore, when modeling BAPO, the reaction byproducts are assumed to have no absorptivity in the emitting range of the UV LED. This eliminates Equation (3) and reduces Equation (4) to the modified Equation (12):(12)$$\frac{\partial I_{j}\left(z,t\right)}{\partial z}=-\left[\varepsilon_{ij}C_{i}\left(z,t\right)+A_{mj}\right]I_{j}$$

Utilizing Equations (2), (5)–(7) and (12), the model was solved for BAPO at 0.1 wt%, 0.5 wt%, and 1.0 wt%, which are equivalent to molar concentrations of 0.00267 mol/L, 0.01329 mol/L, and 0.02645 mol/L, respectively. A continuous, constant UV irradiation was assumed, with an intensity output matching the Altair 75 lamp, at a peak irradiation of 380 mW/cm^2^.

Figure 6a shows the theoretical photoinitiation rates for BAPO at the 1.0 wt% concentration. Note that the peak photoinitiation rate shifted over a 10 s period from a 0.5-mm depth at 0.5 s to a 4.7-mm depth, indicating an initiation wavefront due to photobleaching. The peak initiation rate occurred closest to the irradiated surface, and as photoinitiation progressed into the polymer, the peak rate also decreased. Kenning reported this same trend for BAPO at 0.0268 mol/L concentration and a 93 mW/cm^2^ light intensity, with falling peak-initiation rates at increasing time and depth [49].

The predicted photoinitiation rates at a depth of 3 mm are shown in Figure 6b. As this figure shows, the Kenning optical model predicted the most rapid onset of photoinitiation at the 0.1 wt% BAPO concentration. In contrast, the highest PI concentration of 1.0 wt% BAPO yielded the highest photoinitiation rate. However, at the 1.0 wt% BAPO concentration, peak photoinitiation rate was not achieved until nearly 6 s of continuous irradiation.

### 4.3. Radiometry for 60 s Irradiation

With continuous irradiation of 380 mW/cm^2^ for 60 s, the measured transmitted-light intensity increased with time for BAPO and the BDMB control formulations. As shown in Figure 7, the formulations with the 0.1 wt% PI concentration allowed greater light transmission than the higher tested concentration levels. The BDMB control formulations also consistently allowed more light transmission than BAPO formulations at the same wt% concentration. For the BAPO formulations, the highest transmitted-light intensity at 60 s is 74.0 mW/cm^2^, only 19% light transmission. The peak transmitted-light intensity for the baseline Bis-EMA:PETIA without any PI was 136 mW/cm^2^, or 36% light transmission (see Appendix A). These findings indicate that complete photobleaching had not occurred by 60 s of irradiation even at 0.1 wt% BAPO.

### 4.4. Rate of Conversion and Degree of Conversion for 60 s Irradiation

As shown in Figure 8, DC was calculated from RT-FTIR results over the 60 s irradiation time and converted into a polymerization rate as the change in DC per second.

The polymerization rate in Figure 8 clearly distinguishes the curing behavior of BAPO formulations and the BDMB control. As was predicted by the optical model in Figure 6b, higher concentrations of BAPO resulted in a delayed onset of the peak polymerization rate. The BDMB-control formulations exhibited a less-consistent delay, with the slowest start of polymerization for the lowest concentration of 0.1 wt%. This delay confirms the influence of photobleaching on BAPO-curing behavior during continuous irradiation and the presence of an initiation wavefront created by free-radical generation in response to changing light penetration over time. However, the delay in cure observed by RT-FTIR is higher than the delay in photoinitiation predicted by the optical model. For the 1.0 wt% concentration of BAPO, peak polymerization rate occurred after 20 s, while the optical model predicted peak photoinitiation at 6 s. Most notably, the trend in polymerization rate (see Figure 8) versus predicted photoinitiation rate (see Figure 6b) is the opposite, with the polymerization rate peak decreasing as the PI concentration increases.

As shown in Figure 9, the ultimate DC for each photopolymer formulation was measured 10 min after 60 s of UV irradiation. The average DC was calculated with a minimum of three specimens per formulation. All formulas achieved an average bottom-surface DC ranging between 66% and 76%.

Multiple comparison of means for one-way ANOVA determined that 0.1 wt% and 1.0 wt% BAPO formulations differed significantly, but 0.5 wt% BAPO was not significantly different from 1.0 wt% BAPO in the final DC. The 1.0 wt% BAPO formulation had the lowest average DC, comparable to the 0.1 wt% BDMB formulation. The 0.5 wt% BDMB formulation had the highest average DC but did not differ significantly from the 0.1 wt% BAPO formulation.

### 4.5. FTIR Analysis for Pulse-Irradiation Testing

Specimens from each formulation were cured under high-intensity UV light for an interval of 1.0 s or 3.7 s, equivalent to a dose of 1.4 J/cm^2^ and 5.5 J/cm^2^, respectively. These specimens were irradiated at a higher lamp intensity than the 60-s-irradiation specimens. The lower intensity was used for the 60-s specimens to help improve resolution in the RT-FTIR data by reducing cure rates.

At 10 min post-irradiation, FTIR was used to determine the DC at a 3-mm depth, with the 1.0 s irradiation indicated as ‘Low Dose’ in Figure 10. In contrast, the 3.7-s irradiation is labeled ‘High Dose.’ While all formulations achieved a high DC in the 60-s-irradiation testing, the 0.5 wt% and 1.0 wt% BAPO formulations had <5% DC at the Low Dose, indicating no cure on the bottom (non-irradiated) surface. Even at the higher dose, the 1.0 wt% BAPO formulation was still uncured at a 3-mm depth, while both 0.5 wt% and 0.1 wt% BAPO formulations achieved >50% DC. Among the BDMB controls, only the 0.1 wt% BDMB formulation did not cure at the Low Dose, although the higher concentration formulas only achieved >50% DC at the higher dose. Of the formulas tested, the 0.1 wt% BAPO formulation achieved the highest DC at Low Dose.

### 4.6. Flexural Testing

The pulse-irradiation DC results selected four formulations for further mechanical testing: 0.5 wt% BAPO, 0.1 wt% BAPO, 0.5 wt% BDMB, and 1.0 wt% BDMB. Flexural test specimens were irradiated at the high dose of 5.5 J/cm^2^. Specimens were tested a minimum of 24 h post-irradiation, with the UV-irradiated side facing up in the three-point-bend fixture. The average test results are given in Table 1, while representative stress-strain curves from each dataset are shown in Figure 11.

The test results shown in Table 1 show that the four formulations exhibited similar mechanical properties. By Tukey HSD and multiple comparison of means, none of the groups differ in flexural strength. However, the 0.5 wt% BAPO formula had the lowest flexural modulus, while 0.1 wt% BAPO and 0.5 wt% BDMB had the highest flexural modulus.

The most common feedstock for LFAM printers is acrylonitrile butadiene styrene (ABS), often blended with chopped fibers to improve mechanical properties and reduce warp [99,100,101]. Depending on processing, the reported flexural modulus for neat ABS used in the BAAM printer ranges from 1.87 GPa–2.45 GPa, while flexural strength varies from 58.6 MPa–68.3 MPa [4]. All tested photopolymer formulations in the present study exceeded the flexural strength of ABS by an average of 46–71% and the flexural modulus of ABS by an average of 25–44%. It is important to note that the photopolymer formulations achieved these mechanical properties despite the presence of a cure gradient.

## 5. Discussion

### 5.1. Optical Model

In the present study, the Kenning optical model proved helpful in predicting the delayed onset of photoinitiation with a photobleaching initiator. However, there was an inverse correlation between the predicted photoinitiation rate and the conversion rates measured via RT-FTIR. The trend in the Kenning model is supported experimentally by Christmann et al., albeit for thin-film specimens of a 25-μm thickness and a low light intensity of 10 mW/cm^2^ [26]. They reported that doubling the concentration of TPO in a glassy-acrylate system from 0.5 wt% to 1.0 wt% increased the polymerization rate by 140% and reduced the time interval to peak conversion rate by half [26]. Christmann et al.’s findings support the trends predicted by the Kenning optical model but not the measured conversion rates in the present study. For thin films such as those in Christmann et al.’s study, the light intensity is assumed to be equal throughout the entire specimen, with no light gradient and negligible photofrontal polymerization [50,89]. The difference between the predicted light gradient and the measured light gradient is a likely cause for the discrepancy between the optical model and the measured conversion rate.

### 5.2. Photobleaching in BAPO and PI Concentration Effects

Radiometry and FTIR testing were used to gain insight into photobleaching in BAPO and the resulting photofrontal polymerization that was predicted by the optical model. However, experimental results at 1.0 wt% BAPO indicated almost undetectable light transmission through the 3-mm specimens (see Figure 7). Under pulse irradiation, the 1.0 wt% BAPO formula was effectively uncured at the bottom surface, and even with 60 s irradiation the bottom surface DC was lower than all other formulas. These findings indicate an interfering effect that is partially negating the photobleaching behavior of BAPO at high concentrations at a 3-mm thickness and limiting the reaction at the bottom surface. Note that BDMB, which does not exhibit photobleaching in the UV region of interest, had both a higher DC after 60 s irradiation and higher light transmission for the same concentrations (see Figure 7 and Figure 9). This can be attributed to the lower molar absorptivity of BDMB, which is 2.7 times lower than BAPO at 398 nm. BDMB has greater light transmission and improved through-cure than BAPO for the same concentration, but BAPO is a more efficient PI at 398 nm. While 0.1 wt% BDMB concentration failed to achieve measurable through-cure at 1.0 s of irradiation, the same concentration of BAPO achieved the highest through-cure of any tested BAPO formulation.

Unlike the 1.0 wt% BAPO formula, the 0.1 wt% BAPO formula exhibited a rapid through-thickness cure response. When comparing Low Dose to 60 s of irradiation, the 0.1 wt% BAPO formula had the smallest difference between the two doses of any formula, with a 25.4% DC drop. It is significant that the 0.1 wt% BAPO formula achieved >45% DC at a 3-mm depth even at the 1.0 s Low Dose irradiation. It demonstrates that this formula is not as sensitive to changes in dose as the other formulations.

Of the formulations tested, the 0.1 wt% BAPO concentration proved optimal for a 3-mm-layer thickness. Radiometry testing demonstrated a high level of light transmission, which correlated with the rapid onset of curing at depth as measured by RT-FTIR. The 0.5 wt% BDMB control formula achieved the highest average DC in the high-dose pulse and continuous irradiation. Still, flexural testing showed that the 0.1 wt% BAPO formula achieved comparable mechanical properties even with a slightly lower DC of 55.5% vs. 60.4% for 0.5 wt% BDMB as measured at a 3-mm depth.

A key difference between the 1.0 wt% BAPO formula and the 0.1 wt% BAPO formula is the initial light absorption. The higher concentration of BAPO absorbs 10× as much light as the 0.1 wt% BAPO formula, limiting light penetration and the through-thickness curing reaction. In a theoretical uninhibited-photobleaching reaction, both formulas would achieve comparable light penetration over time. However, for the Bis-EMA:PETIA photopolymer in this study, experimental results confirm that the lower light absorption of low BAPO concentrations has a stronger influence on improving the through-thickness cure response in 3 mm than the photobleaching behavior, in both pulse and continuous irradiation.

### 5.3. Effect of Radical Trapping

While complete cure occurs when all of the double bonds have been converted (100% DC), phenomena such as the cage effect can trap unreacted free radicals within a growing-polymer network and limit ultimate DC [39,72]. Ultimate DC below 100% is typical for glassy-acrylate polymers, as radical occlusion can surpass bimolecular termination by a ratio of 4 to 1 during vitrification [96]. Trapped free radicals in a glassy system reportedly exceed active (e.g., mobile) free radicals at as low as 36% DC [74]. Bis-EMA:PETIA is a glassy polymer system as indicated by its (meth)acrylate chemistry, reported T_g_ of 150 °C, high stiffness (tensile modulus > 2.0 GPa), and brittle failure [24]. As would be expected for a glassy system, the highest average DC was 76.0%, as measured at the bottom surface of the 0.5 wt% BDMB formulation after 60 s irradiation.

As a glassy-polymer network vitrifies during curing, trapped free radicals can become dislocated and eventually react through diffusion, or they can recombine back into a PI molecule (see Figure 12). Eibel et al. considered recombination the most likely side reaction for initiator radicals, noting that this becomes insignificant at a high monomer concentration [30]. However, when the monomer concentration drops and a polymer network vitrifies, the likelihood of in-cage recombination increases [102]. In addition to reducing the effective quantum yield [103], in-cage recombination can counteract photobleaching as recombined PI molecules absorb light in vitrified regions. This phenomenon is not factored into the optical model, which does not predict DC or the likelihood of free-radical trapping. The radiometry data support the hypothesis of free-radical trapping and in-cage recombination, with transmitted-light intensity 100 times lower for 1.0 wt% BAPO than for 0.1 wt% BAPO after 60 s of irradiation. The delayed curing reaction observed in RT-FTIR provides evidence of photobleaching. Still, the lower reaction rate and low light transmission at higher PI concentrations may indicate a high residual PI content trapped within the cured polymer. This trend is confirmed by Christmann et al., who reported that increased PI concentration increases radical trapping by occlusion in the glassy-3D-polymer networks [26]. The optical model from Kenning and Miller does not account for this behavior directly, although it could be considered as ‘partial photobleaching’, where $\varepsilon_{pj}\neq 0$ as described by Hayki et al. [50].

### 5.4. Utility of Optical Properties in Formulation Pre-Screening

For applications such as LFAM where a short irradiation time (<5 s) is desirable, another potential pre-screening tool is the predicted light intensity through the polymer at t=0, or the starting point of irradiation. As shown in Figure 13, the Kenning optical model predicts light intensity > 0 throughout the entire 3-mm polymer thickness, for formulations with 0.1 wt% BAPO, 0.1 wt% BDMB, and 0.5 wt% BDMB. Two of these formulas proved the best suited for rapid cure during pulse irradiation testing, but the 0.1 wt% BDMB formulation had the lowest DC of the tested BDMB formulations. Therefore, this limited use of the Kenning optical model may be a helpful tool for screening out formulations with too much PI, but other approaches such as FTIR are required to screen for formulations with PI levels too low for complete cure.

## 6. Conclusions

This study investigated light transmission and through-thickness cure behavior in (meth)acrylate photopolymer formulations, with the photobleaching initiator BAPO cured at a 3-mm thickness. Formulations with BDMB were also tested as a control. For the first time reported in the literature, a theoretical optical model was employed to predict photoinitiation rates, with BAPO as a potential screening tool for LFAM formulations, which was then tested against experimental results. The model predicted an increase in photoinitiation rate at higher initiator concentrations, accompanied by a temporal delay in the onset of peak initiation due to photobleaching. BAPO formulations were prepared and irradiated for 60 s for comparison against the optical model. At a 3-mm depth, experimental results indicate increasing BAPO concentration from 0.1 wt% to 1.0 wt% delayed peak conversion time by over 18 s, confirming photobleaching behavior. Increased BAPO concentration also reduced the peak conversion rate by 75% over the same range, an inverse trend to the optical model predictions. This reduction may be attributed to in-cage recombination of photoinitiators in the vitrified polymer, which increases light attenuation. This hypothesis is supported by radiometry results, which show that peak light transmission at 0.1 wt% BAPO is 50% lower than the light transmission through the polymer without any photoinitiator.

Although predicted photoinitiation rates from optical modeling did not match measured polymerization rates, the optical model does offer some utility as a screening tool for establishing an upper limit initiator concentration based on optical density. The tested formulations that cured most rapidly through a 3-mm thickness also had predicted light transmission > 0 throughout the thickness at t = 0. These potential screening tools confirm the importance of balancing PI concentration with PI light absorption to allow the greatest initial light transmission.

For layer thicknesses of 3 mm, this study found that an 80:20 Bis-EMA:PETIA blend achieved the most rapid through-thickness cure response with 0.1 wt% BAPO as the PI concentration. For the same 3.7 s irradiation, 0.1 wt% BAPO and 0.5 wt% BDMB formulations achieved comparable material properties, with an average flexural strength >100 MPa and flexural modulus of 3.1 GPa, higher than ABS and well-suited for LFAM applications. While photobleaching in BAPO appeared to provide minimal benefit under simulated LFAM conditions, the higher molar absorptivity of BAPO relative to BDMB enabled comparable performance at lower concentrations. Since PI cost can significantly impact overall photopolymer-formulation cost, this means that BAPO is a more cost-effective option than BDMB for the tested curing conditions. The 0.1 wt% BAPO concentration also had the highest through-cure DC with 1 s irradiation, demonstrating that it is an effective PI for methods such as LFAM that must rely on minimal UV exposure times to maximize material throughput rates. Based on the findings of the present study, the 0.1 wt% BAPO formulation is recommended over the previously reported 1.0 wt% BDMB formulation for potential LFAM applications.

## Figures and Tables

**Figure 1 polymers-14-02708-f001:**
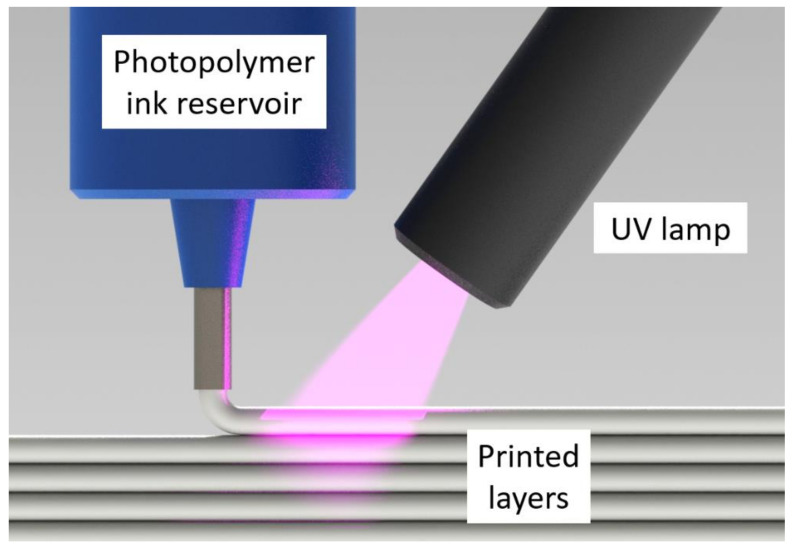
Schematic depicting the UV-DIW printing process, in which a photopolymer ink is deposited and rapidly cured in situ.

**Figure 2 polymers-14-02708-f002:**
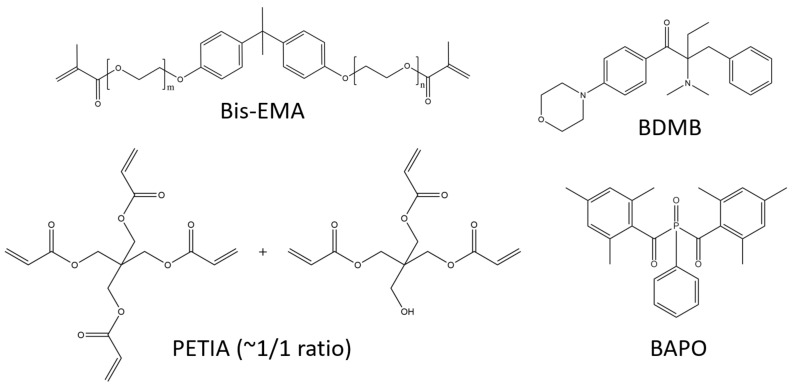
Primary oligomer Bis-EMA is di-functional while PETIA has tri- and tetra-functionality. BAPO is a common photobleaching PI, while BDMB is highly reactive.

**Figure 3 polymers-14-02708-f003:**
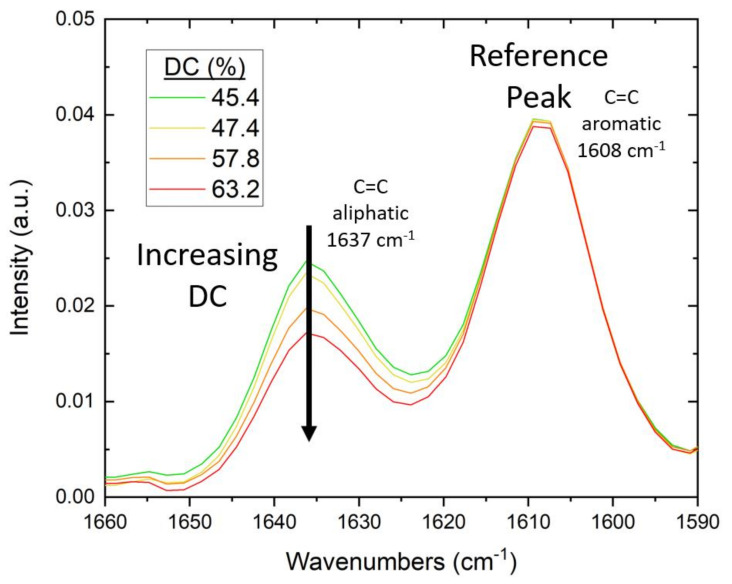
Example of change in FTIR spectra with increasing DC of Bis-EMA photopolymers. As the (meth)acrylate pendant groups react during curing, the intensity of the aliphatic C=C peak at 1637 cm^−1^ decreases.

**Figure 4 polymers-14-02708-f004:**
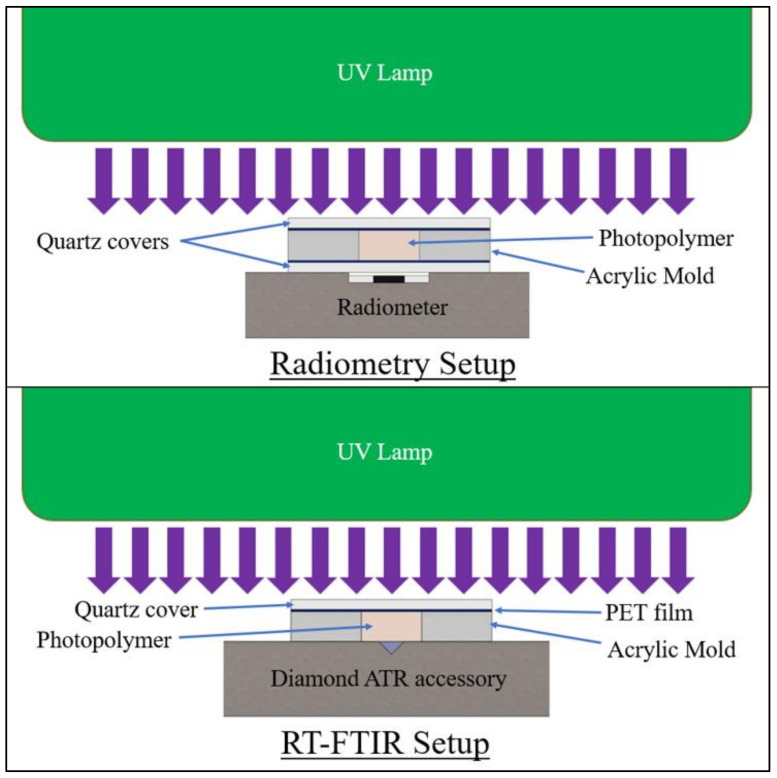
Specimen setup for 60 s irradiation testing at 3-mm specimen thickness.

**Figure 5 polymers-14-02708-f005:**
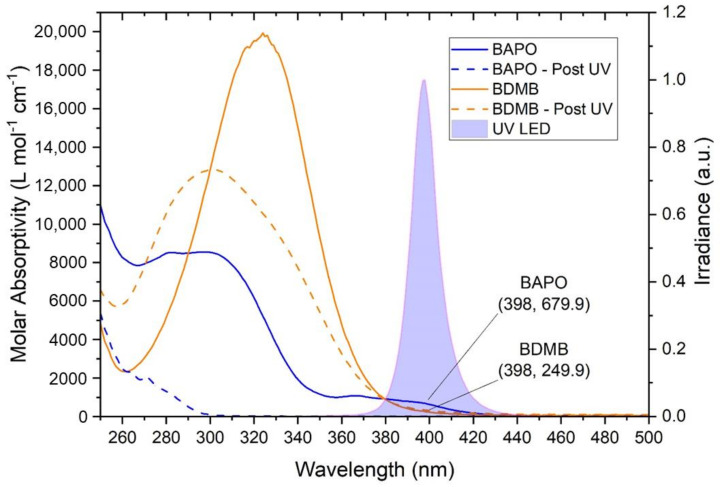
The molar absorptivity of BAPO and BDMB, with the reaction byproduct molar absorptivity indicated by ‘Post UV.’ The absorptivity is compared with the spectral output of an Altair 75 lamp, with the measured spectral output indicating a peak UV output at 398 nm.

**Figure 6 polymers-14-02708-f006:**
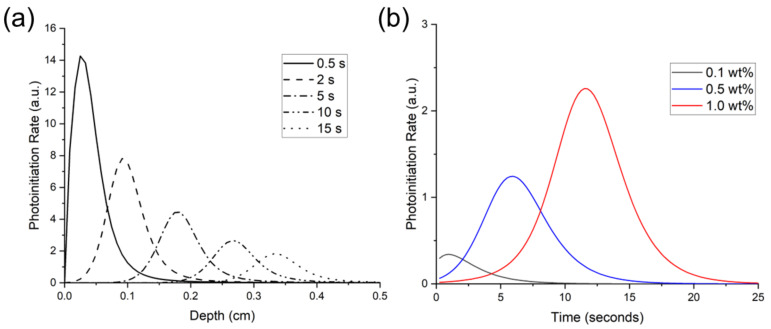
Predicted photoinitiation rate for BAPO system in an 80:20 blend of Bis-EMA:PETIA. (**a**) depicts the temporal and spatial evolution of photoinitiation as the BAPO photobleaches at 1.0 wt% concentration, and (**b**) depicts the rate over time at a 3-mm depth with varying BAPO wt%.

**Figure 7 polymers-14-02708-f007:**
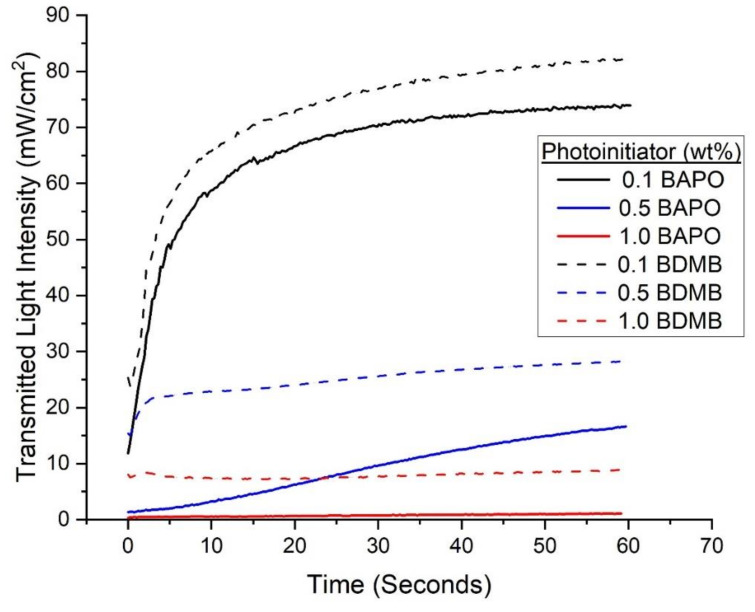
Experimental values for transmitted-light intensity at varying concentrations of PI for BAPO and BDMB through a 3-mm thickness of Bis-EMA/PETIA polymer.

**Figure 8 polymers-14-02708-f008:**
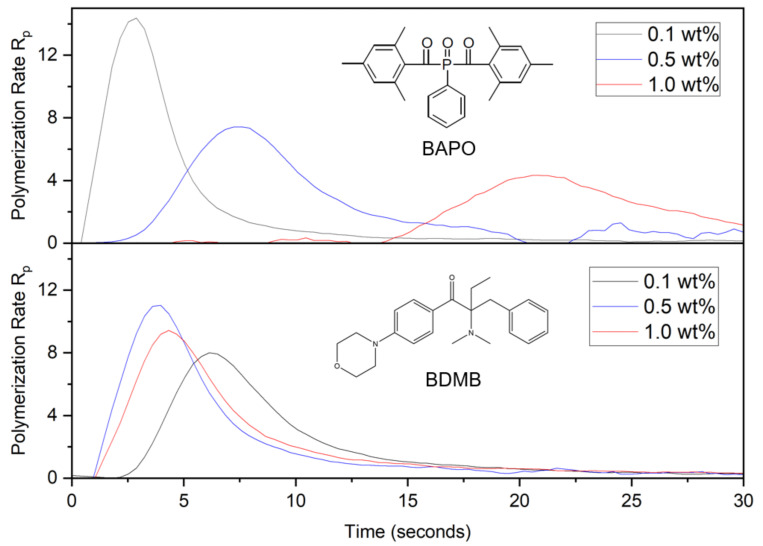
Polymerization rate, or change in DC per second, for BAPO and BDMB at various concentrations, measured at a 3-mm depth under continuous irradiation of 380 mW/cm^2^ UV light.

**Figure 9 polymers-14-02708-f009:**
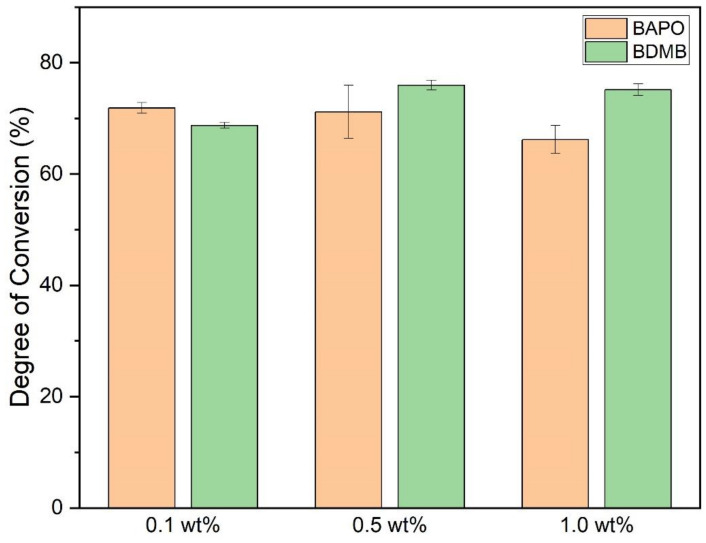
DC post-60-s UV irradiation, measured at a 3-mm polymer depth. Error bars indicate +/− one standard deviation.

**Figure 10 polymers-14-02708-f010:**
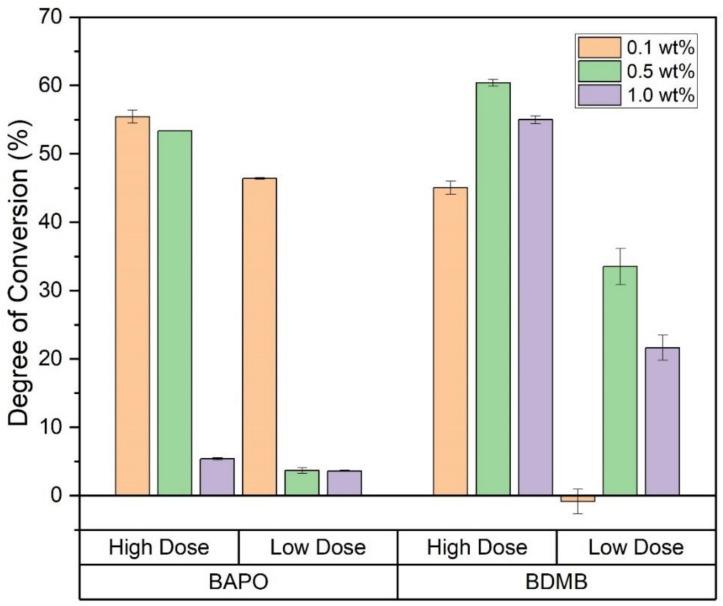
DC for all six formulas as measured at a 3-mm depth, with Low Dose irradiation of 1.4 J/cm^2^ or High Dose irradiation of 5.5 J/cm^2^. Error bars indicate +/− one standard deviation.

**Figure 11 polymers-14-02708-f011:**
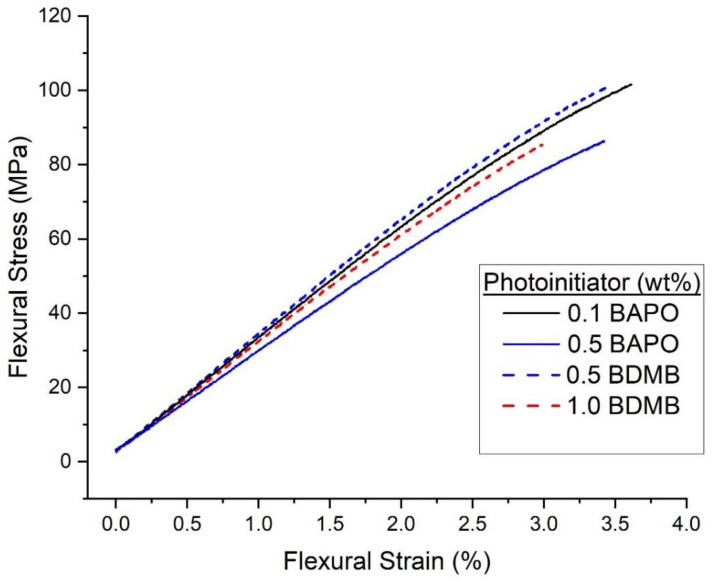
Representative flexural stress-flexural strain curves for the neat resin formulas irradiated under high intensity (1600 mW/cm^2^) for 3.7 s.

**Figure 12 polymers-14-02708-f012:**
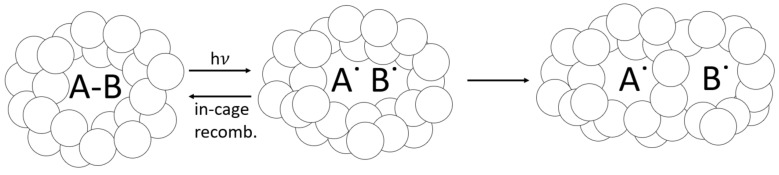
Cage effect, which can apply to occluded PI molecules. As PI absorbs light and reacts, its reaction byproducts can become trapped within the tightly packed vitrified-polymer network, which will either recombine or dissociate. Some free radicals can still react via diffusion, even in the glassy polymer.

**Figure 13 polymers-14-02708-f013:**
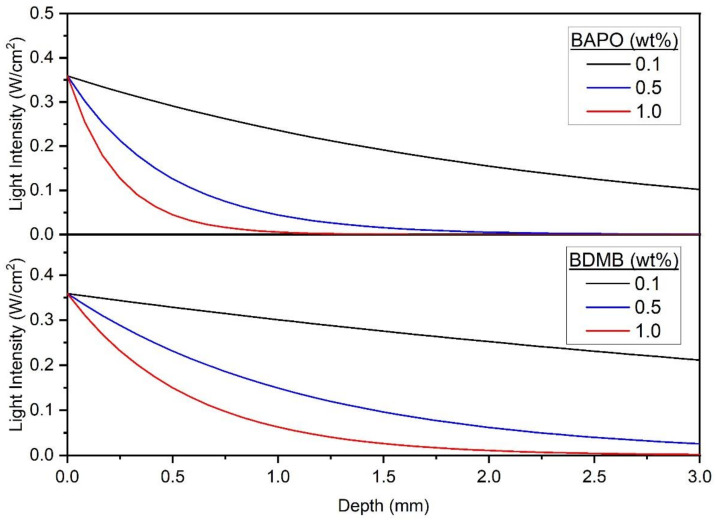
Predicted transmitted-light intensity at t = 0 across the 3.0 mm polymer depth, at 398 nm wavelength and an incident-light intensity of 380 mW/cm^2^.

**Table 1 polymers-14-02708-t001:** Average flexural properties from three-point-bend testing, [x] indicates standard deviation.

	BAPO		BDMB	
	0.1 wt%	0.5 wt%	0.5 wt%	1.0 wt%
Flexural Strength (MPa)	108.7 [11.2]	94.6 [9.7]	101.4 [10.1]	95.4 [14.9]
Flexural Modulus (GPa)	3.1 [0.0]	2.7 [0.1]	3.1 [0.1]	2.9 [0.1]
Failure Strain (%)	4.0 [0.8]	4.1 [0.8]	3.6 [0.6]	3.7 [0.9]

## Data Availability

Not applicable.

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
