# Peer review of "Photoinitiator Selection and Concentration in Photopolymer Formulations towards Large-Format Additive Manufacturing"

_polymers, 2022, doi:10.3390/polym14132708_

Round 1

Reviewer 1 Report

Dr. Vaidya and coworkers reported an interesting work demonstrates the utility of optical modeling as a potential screening tool for new photopolymer formulations. The paper is well written, and the results support their conclusion. The topic about photo 3D printing is also very hot currently. However, the following questions must be addressed before publication.

  1. The advantages of “deep cure” enabled by photobleached PIs are not very clear to me. The authors could add a couple of sentences of discussion in the introduction.
  2. Do the authors think this photobleaching PIs strategy has limitations? In my opinion, this method is suitable for DIW but not suitable for SLA or DLP printing, as the photobleaching will cause the inefficient polymerization in the latter two cases. If so, the authors should add some discussion somewhere appropriate.
  3. Some recent important example of nanomaterials photoinitiated polymerization/3D printing are missing (https://doi.org/10.1039/D1PY00705J; https://doi.org/10.1021/acsmacrolett.0c00232). Please cite them somewhere appropriate in the introduction.
  4. How did authors use FTIR to get monomer conversion is not clear. Please revise it in section 4 accordingly and give the equation.
  5. The authors used vinyl ester to describe the monomer in Figure 2, which is not accurate. Please see the vinyl ester definition here (https://en.wikipedia.org/wiki/Vinyl_ester). “Acrylates” are better.
  6. How did authors get “BDMB has a molar absorptivity of 249.9 L/mol-cm while BAPO is 340 679.9 L/mol-cm” ? Not reference or data support here.
  7. Many typos in the paper please correct them accordingly. For example, page 13, “cm-1”.
  8. The photobleaching by products/reaction of BAPO could be shown in supporting information.

Author Response

The authors would like to thank the reviewer for their insightful comments and willingness to review and help improve this manuscript.  We have added significant clarifying language to the manuscript to help better convey some of the key concepts such as deep cure and FTIR degree of conversion testing.  Please see a detailed response in the attached file.

Reviewer 1 - All changes are highlighted in red in the revised manuscript

Reviewer Comment

Authors Response

1

The advantages of “deep cure” enabled by photobleached PIs are not very clear to me. The authors could add a couple of sentences of discussion in the introduction.

The authors have added further explanation of photobleaching in the introduction section, with a comparison against other hybrid approaches.

2

Do the authors think this photobleaching PIs strategy has limitations? In my opinion, this method is suitable for DIW but not suitable for SLA or DLP printing, as the photobleaching will cause the inefficient polymerization in the latter two cases. If so, the authors should add some discussion somewhere appropriate.

The authors have added further explanation of photobleaching PIs in the introduction, including some consideration of SLA and DLP printing.  BAPO is a common PI for SLA and DLP photopolymers, in part because it becomes optically transparent upon prolonged UV exposure.  BAPO and other phosphine oxides also have high molar absorptivity in the UVA region, overlapping with the 405 nm light sources often used in SLA.  The authors have added further comments on the potential limitations of photobleaching PIs in the Discussion section.

3

Some recent important example of nanomaterials photoinitiated polymerization/3D printing are missing (https://doi.org/10.1039/D1PY00705J; https://doi.org/10.1021/acsmacrolett.0c00232). Please cite them somewhere appropriate in the introduction.

The authors have covered the literature adequately and cannot promote a single author.  The suggested publications do not appear to contribute to the overall narrative of deep cure systems, photobleaching photoinitiators, UV curing in the UVA region and photocuring in additive manufacturing.

4

How did authors use FTIR to get monomer conversion is not clear. Please revise it in section 4 accordingly and give the equation.

The method is described in detail in a previous paper, which the authors have now referenced and summarized in Section 3.5

5

The authors used vinyl ester to describe the monomer in Figure 2, which is not accurate. Please see the vinyl ester definition here (https://en.wikipedia.org/wiki/Vinyl_ester). “Acrylates” are better.

The authors have added clarifying language in the Materials and Methods section, and refer to Bis-EMA as a methacrylate and PETIA as an acrylate. All references to vinyl ester have been replaced with the more descriptive term (meth)acrylate, indicating that both methacrylates and acrylates were used in the study.   

6

How did authors get “BDMB has a molar absorptivity of 249.9 L/mol-cm while BAPO is 340 679.9 L/mol-cm” ? Not reference or data support here.

The authors have combined the figures showing molar absorptivity and added data call-outs indicating the specific data points at 398 nm referenced in the manuscript.  This can now be found as Figure 5.

7

Many typos in the paper please correct them accordingly. For example, page 13, “cm-1”.

The authors checked units against the guidelines provided by NIST (https://physics.nist.gov/cuu/Units/checklist.html) and have corrected any reference of 1/cm to cm-1 and L/mol-cm to L mol-1 cm-1.  We also conducted a thorough review of the manuscript for typos, please advise on any other specific instances.

8

The photobleaching by products/reaction of BAPO could be shown in supporting information.

A figure and brief description of the BAPO reaction byproducts has been added to the supplemental information as Figure S1.

Reviewer 2 Report

In this work, Stiles and colleagues studied the use of BAPO as a photoinitiator for 80:20 Bis-EMA:PETIA blend cross-linking.  BDMB was used as the control.  The authors accompanied the reactions using UV/Vis and FTIR (short and long-irradiation studies were conducted) and evaluated the mechanical properties of the composites.

In the end they show that BAPO’s performance is not inferior to BDMB.  I believe this study is interesting, well conducted and relevant for future materials science applications.  However, some key aspects need to be addressed before publication.  As such, I ask the editor to return the manuscript with Major Revisions.  Please consider the following:

  1. I believe the manuscript should be re-arranged to fit a standard paper format. This should include:

1.1. Consider organizing a proper results and discussion section with no clear intermediate “discussion” sections (e.g. page 10) and/or separate general discussion (e.g. Pages 17 to 19) and a more fluid writing style that does not resemble that of a report.

1.2. I advise the authors to summarize (as best as possible) and include sections 2 and 3 in the introduction section.

1.3.  Please check if the unit system used conforms with SI norms.  For example, L/mol-cm (line 341) and “1/cm” (Figure 7) do not.

1.4. Present a proper and individualized “statistical analysis” sub-section in the materials and methods section.

  1. I believe there are too many figures in this manuscript. I advise the authors to address this issue by constructing panels where two or more current figures are merged on.
  2. In page 7, line 294, can the authors explain what “pph” stands for?
  3. Regarding the FTIR protocol of section 4.4., please explain the following:

4.1. How many scans were performed in the real-time FTIR?  Is this the “gain”?.  Please elaborate on this and include the necessary information in the text if necessary.

4.2. Why the authors accumulated 64 scans in the 10-minutes post-radiation?  I believe too many accumulations might lead to information loss and biased results, especially for low intensity peaks.

  1. In section 4.6., you indicate that the number of specimens used was more than 6 per sample. I consider this not to be appropriate for the correct evaluation of the results. Please use the same number of specimens for your calculations.

5.1. Explain how each of the parameters were calculated in the materials and methods section.

5.2. Present representative stress-strain curves for the materials.

5.3. Include calculation for the mechanical properties of the composites cured for 60 s.

  1. Please elaborate more on the photobleaching effect and how did it interfered with the curing with BAPO.
  2. In phrases 413-415, the author indicate that certain FTIR peaks were used as reference to evaluate the degree of conversion. Please consider the following:

7.1. The formula for the calculation of the degree of conversion, along with the others presented throughout the text, should be moved to the materials and methods section.

7.2. Why were the aforementioned peaks used for your calculation?  Please present representative FTIR spectra in your figures for better data visualization.

7.3. Please consider remaking your calculations using FTIR spectra acquired with less scan accumulations (e.g. 8).

  1. The statistical analysis presented in phrases 444-448 should be replicated in the respective figure.
  2. The cage effect described by the authors is an interesting hypothesis to explain the deficiency in curing. Please perform DSC (2-3 cycles, no pre-heating stage, 5 or 10 ºC per min), XRD and/or other appropriate chemical analysis techniques on the composites to fully understand phase separation.
  3. In figure 14, the error bar is missing for the green bar in the BAPO-High Dose part of the figure.
  4. Please explain the information contained in phrase 493-494 and why the authors proceeded with such protocol instead of using the same lamp intensity.
  5. I believe the text lacks a final evaluation of the performance of BAPO. Please further elaborate on this and explain what advantages this formulation has for industrial applications.

Author Response

The authors would like to thank the reviewer for their detailed comments, and for their efforts to help improve this manuscript.  We have significantly restructured the paper to have a more clearly delineated Results section and Discussion section, and have worked to improve the overall narrative flow.  We have also combined several figures and added new figures where requested.  Please see a detailed response in the attached file.

Reviewer 2 - Where possible, all changes are highlighted in blue in the manuscript

Reviewer Comments:

Authors Response:

1

I believe the manuscript should be re-arranged to fit a standard paper format. This should include:

1.1

Consider organizing a proper results and discussion section with no clear intermediate “discussion” sections (e.g. page 10) and/or separate general discussion (e.g. Pages 17 to 19) and a more fluid writing style that does not resemble that of a report.

The authors have reorganized the Discussion section to incorporate the intermediate discussion sections.  We have correspondingly trimmed content from the Results section and worked to achieve a more narrative flow throughout the Discussion section.

1.2

I advise the authors to summarize (as best as possible) and include sections 2 and 3 in the introduction section.

The authors have considered this section, and feel that the concluding paragraphs of the Introduction and Literature Review sections suffice as a means of summarizing the study and the key objectives.

1.3

Please check if the unit system used conforms with SI norms.  For example, L/mol-cm (line 341) and “1/cm” (Figure 7) do not.

The authors checked these units against the guidelines provided by NIST (https://physics.nist.gov/cuu/Units/checklist.html) and have corrected any reference of 1/cm to cm-1 and L/mol-cm to L mol-1 cm-1

1.4

Present a proper and individualized “statistical analysis” sub-section in the materials and methods section.

The authors have considered this suggestion, and have added the new subsection 3.8.

1

I believe there are too many figures in this manuscript. I advise the authors to address this issue by constructing panels where two or more current figures are merged on.

Figures 2 and 3 were combined to show all materials in a single figure, Figures 5 and 6 were combined to show all photoinitiator absorptivity data, and Figures 8 and 9 were combined to show both aspects of the predicted photoinitiation rates.  Figure 7 and the accompanying description was also removed, as a more complete figure and discussion already existed in the Supplemental Materials.

2

In page 7, line 294, can the authors explain what “pph” stands for?

This was clarified to indicate the 80:20 ratio referenced later in the paper, or 80 parts Bis-EMA to 20 parts PETIA by weight.

3

Regarding the FTIR protocol of section 4.4., please explain the following:

4.1

How many scans were performed in the real-time FTIR?  Is this the “gain”?.  Please elaborate on this and include the necessary information in the text if necessary.

The authors have expanded this section in the text.  The real-time FTIR used a series of individual scans, with each scan taking 1.3 s.  Scans were taken continuously every 1.3 s for 10 minutes.

4.2

Why the authors accumulated 64 scans in the 10-minutes post-radiation?  I believe too many accumulations might lead to information loss and biased results, especially for low intensity peaks.

The authors selected 64 scans as it is the method reported for many instances of FTIR degree of conversion measurements in photopolymers.  These scans are completed in approximately 83 s, at which point the software automatically combines the data into a single spectrum.  FTIR testing of the 0.1% BAPO formulation showed that the DC changed by      < 0.06% per minute between 10 minutes and 1 hour post irradiation, and therefore the potential change in DC during the 64 scans at 10 minutes post irradiation can be considered negligible.

1

In section 4.6., you indicate that the number of specimens used was more than 6 per sample. I consider this not to be appropriate for the correct evaluation of the results. Please use the same number of specimens for your calculations.

For flexural testing, the authors have carefully adhered to the guidelines of ASTM D790, which specifies a minimum of 5 specimens for each test condition.  Each data set contained 6 to 9 specimens, with some variability due to the molding process. Per the reviewer's suggestion, we have removed specimens from some of the data sets such that only the first 6 specimens from each data set remain. The recalculated mechanical properties are shown in the updated Table 1.  The properties did not significantly change with this adjustment, and the conclusions and statistical analysis remain valid.

5.1

Explain how each of the parameters were calculated in the materials and methods section.

This comment is unclear, please indicate which parameters need further explanation.

5.2

Present representative stress-strain curves for the materials.

Figure 11 has been added showing the stress-strain curves for the test specimens that were the closest to the average values for each tested data set.

5.3

Include calculation for the mechanical properties of the composites cured for 60 s.

The 60 s irradiation time at reduced intensity was primarily utilized as a means of capturing the polymerization rate via FTIR and for comparison against the optical model.  For the desired end application of AM, the mechanical properties for specimens cured under a shorter irradiation at higher lamp intensity are the most relevant.  At this point, flexural specimens at the 60 s irradiation time have not been tested, and this project does not have funding available for further testing.

1

Please elaborate more on the photobleaching effect and how did it interfered with the curing with BAPO.

The authors have restructured the Discussion section to better explain the connection between the cage effect and the observed BAPO behavior.  The photobleaching effect is not what interfered with the curing of BAPO, but rather the cage effect disrupted photobleaching and the high molar absorptivity of residual unreacted BAPO is what blocked light transmission deeper into the polymer and restricted through-thickness curing.

2

In phrases 413-415, the author indicate that certain FTIR peaks were used as reference to evaluate the degree of conversion. Please consider the following:

7.1

The formula for the calculation of the degree of conversion, along with the others presented throughout the text, should be moved to the materials and methods section.

A more detailed explanation of FTIR degree of conversion calculations is now included in section 3.5.  The optical model methods section has also been moved to the materials and methods section and is now section 3.3.

7.2

Why were the aforementioned peaks used for your calculation?  Please present representative FTIR spectra in your figures for better data visualization.

The peaks for FTIR calculation are now shown in Figure 3 in section 3.5.

7.3

Please consider remaking your calculations using FTIR spectra acquired with less scan accumulations (e.g. 8).

The authors have considered this suggestion, but feel strongly that 64 scans should be utilized when possible to have the closest comparison against the methods used by other authors studying photocuring. 

1

The statistical analysis presented in phrases 444-448 should be replicated in the respective figure.

An in-depth review of the cage effect falls outside of the scope of the manuscript.  The purpose of the figure indicated is to provide a broad cartoon of the cage effect and in-cage recombination, and for greater depth we will defer to the authors whose work is cited in these lines.

2

The cage effect described by the authors is an interesting hypothesis to explain the deficiency in curing. Please perform DSC (2-3 cycles, no pre-heating stage, 5 or 10 ºC per min), XRD and/or other appropriate chemical analysis techniques on the composites to fully understand phase separation.

The authors appreciate this comment and the suggested experiments to validate the cage effect hypothesis.  While a lengthy examination of this effect falls outside of the scope of the present paper, we have conducted a series of preliminary DMA and FTIR tests on various thermal post cure conditions that confirm the cage effect under ambient UV curing.  A post cure of 200 C was found to increase the DC to 98%, which supports the hypothesis that restored polymer chain mobility at temperatures above Tg allows trapped free radicals to drive conversion higher.  We plan to publish these results as a separate manuscript that is primarily focused on the cage effect and combined UV+thermal treatments in Bis-EMA photopolymers.

3

In figure 14, the error bar is missing for the green bar in the BAPO-High Dose part of the figure.

The error for those measurements was 0.026%, and the error was so small that the error bar is not visible

4

Please explain the information contained in phrase 493-494 and why the authors proceeded with such protocol instead of using the same lamp intensity.

The authors have added in the explanation in the materials and methods section 3.6.  The short irradiation specimens were cured using the highest possible light intensity for the lamp system to enable the fastest possible curing reaction.  In the 60 s irradiation testing, a lower intensity had to be used to enable sufficient data capture during real-time FTIR to experimentally determine the change in Rp over time. The authors found that, at higher intensity, the reaction occurred too quickly to capture using the FTIR equipment available. 

5

I believe the text lacks a final evaluation of the performance of BAPO. Please further elaborate on this and explain what advantages this formulation has for industrial applications.

The authors have added a more focused discussion of BAPO in the Discussion section, and provided some additional evaluation of BAPO in the Conclusion.

Reviewer 3 Report

Accept in present form.

Author Response

The authors would like to thank the reviewer for their response and are pleased that you found the manuscript acceptable in its original form.  Per the other reviewer's comments, we have adjusted the formatting somewhat to improve the narrative flow.  We have also added a deeper explanation of FTIR for the degree of conversion calculation and of the role of photobleaching in deep cure systems.

Round 2

Reviewer 1 Report

There is some improvements but away from enough and more improvement is required. Most of problems in the paper are either misleading conceptual mistakes or lacking of references.  The following questions must be addressed:

1. I do not agree with what authors claimed:" Photoinitiation commences when a photoinitiator (PI) molecule absorbs light and forms free radicals." How about photoinitiation of cationic and thiol ene polymerization? Any free radicals? Besides, how about charge transfer from PI to coinitiator? This case the PI do not form free radicals but coinitiators do.

2.  Page 4,  the authors talked about the "Thermal initiators and/or cationic curing chemistries have both been employed for a deep cure and have been used in combination with free-radical curing PIs." and then followed by UCNP initiated 3D printing, and then come back to cationic chemistries. It is a very confusing and no logic. What is UCNP photopolymerization to do with thermal and cationic process? It is a totally different reaction mechanism! Light from upconversion initiated the reaction and has nothing to do with authors' discussion.

3. The authors claimed that " Cationic chemistries are generally more expensive than free-radical curing chemistries" and cited a reference in 2010.  Maybe it was right in 2010, but with the development of living photo cationic polymerization, the cationic polymerization is more efficient and lower the cost. If the authors insist this claim, please find a reference in recent five years, and give us how much less it costs and why. Is this because of the polymerization conversion, photoinitiator price or other reasons?

4. A lot of claim are lacking of references.

a. "Conventional free-radical PIs continue absorbing light after reacting, restricting light penetration throughout the curing reaction." 

b. "To overcome the limited light penetration of some photopolymer formulations, several hybrid curing chemistries have been developed for curing thicker polymers (e.g. greater than 0.5 mm)."

c.  "The latter are ideal for LFAM because they are more compact and can cycle rapidly without a ‘warm-up’ period to achieve peak output intensity."

d."  Phosphine oxides such as TPO, TPO-L, and BAPO all absorb light in the UVA range and are photobleaching, making them strong candidates.."

e. "Several optical models have been proposed using experimental inputs such as PI molar absorptivity, polymer absorptivity, PI concentration, and incident light intensity." Besides, the models should be listed right after this sentence or the flow does not read well.

5. Figure 3 was not presented in a scientific way. I do not understand what each curves represents for. The curves color is the same and what is the difference (the reaction time) should be presented.

6. According to the mdpi guide ( https://www.mdpi.com/reviewers#Review_Report), most of the cited references are better within 5 years. For the author's case, only 11 of 69 references is within 5 years. Most of them are the 1990s or 2000s papers. The authors should cite more recent papers in photo 3D printing or replace some old papers.

7. Any reference for Figure S1?

8. The novelty of this work should be better presented. The photobleaching PIs have been reported a lot before as the authors wrote in their introduction. What is new of this BAPO PIs should be discussed.

9. Following question 8, the authors could make a table and discuss about the different PIs (some examples just fine) in 3D printing with their work in different parameter (bleaching or not, absorption, etc.) and polymerization and 3D printing results. This could help better solve the question 8.

Author Response

The authors would like to thank the reviewer for their guidance in improving this manuscript.  Per the reviewer feedback, we have significantly restructured a portion of the Introduction and now have 57 out of 103 references from publications within the past 5 years.  For a detailed response, please see the attached document.

Reviewer 1 - Where possible, all changes are highlighted in red in the manuscript

Reviewer Comment

Authors Response

1

 I do not agree with what authors claimed:" Photoinitiation commences when a photoinitiator (PI) molecule absorbs light and forms free radicals." How about photoinitiation of cationic and thiol ene polymerization? Any free radicals? Besides, how about charge transfer from PI to coinitiator? This case the PI do not form free radicals but coinitiators do.

The authors have adjusted this claim (lines 73-77) to more broadly include some of the other initiating species.  As the present study considers only conventional free-radical curing photoinitiators in detail, the bulk of the paragraph is primarily focused on that curing mechanism.  Further consideration is given to alternative chemistries later in the introduction (lines 133-169).

2

Page 4,  the authors talked about the "Thermal initiators and/or cationic curing chemistries have both been employed for a deep cure and have been used in combination with free-radical curing PIs." and then followed by UCNP initiated 3D printing, and then come back to cationic chemistries. It is a very confusing and no logic. What is UCNP photopolymerization to do with thermal and cationic process? It is a totally different reaction mechanism! Light from upconversion initiated the reaction and has nothing to do with authors' discussion.

The authors have extensively restructured this section (lines 98-181) to more fully discuss several methods for improving depth of cure in photopolymers, separating each method into a different category.  We have also included a brief discussion of the RAFT process (lines 157-169), and would like to thank the reviewer for directing us to the papers mentioned in the previous revision notes.  The RAFT process has been demonstrated to improve cure response in 4 mm thick dental composites, and may be of interest as a topic of future study in LFAM applications.

3

The authors claimed that " Cationic chemistries are generally more expensive than free-radical curing chemistries" and cited a reference in 2010.  Maybe it was right in 2010, but with the development of living photo cationic polymerization, the cationic polymerization is more efficient and lower the cost. If the authors insist this claim, please find a reference in recent five years, and give us how much less it costs and why. Is this because of the polymerization conversion, photoinitiator price or other reasons?

This section has been extensively restructured (lines 133-156), and this claim has been removed.  A key objective of this paper was to study Bis-EMA/PETIA photopolymers more thoroughly, as they were previously identified as a system which could achieve a high Tg at a low formulation cost suitable for potential LFAM applications.  The authors added a more salient argument against the use of hybrid free-radical/cationic chemistries, that the addition of epoxy to the Bis-EMA/PETIA monomers could alter the cured polymer properties, making a direct comparison against the BDMB control more difficult (see lines 174-177) .

4

A lot of claim are lacking of references:

The authors have added references to all of the instances noted by the reviewer, as follows:

a

"Conventional free-radical PIs continue absorbing light after reacting, restricting light penetration throughout the curing reaction." 

The authors have added relevant references to support this claim (see lines 112-114).

b

"To overcome the limited light penetration of some photopolymer formulations, several hybrid curing chemistries have been developed for curing thicker polymers (e.g. greater than 0.5 mm)."

The authors have rewritten this sentence and added references to support the claim (see lines 98-100).

c

"The latter are ideal for LFAM because they are more compact and can cycle rapidly without a ‘warm-up’ period to achieve peak output intensity."

The authors have added relevant references to support this claim (see lines 216-218).

d

"Phosphine oxides such as TPO, TPO-L, and BAPO all absorb light in the UVA range and are photobleaching, making them strong candidates.."

The authors have added relevant references to support this claim (see lines 219-221).

e

"Several optical models have been proposed using experimental inputs such as PI molar absorptivity, polymer absorptivity, PI concentration, and incident light intensity." Besides, the models should be listed right after this sentence or the flow does not read well.

The authors have restructured the first few sentences of this paragraph to introduce the models at an earlier point, with their respective references.  The potential utility of the models is still included in these opening sentences, but in a more concise form (see lines 238-242).

5

Figure 3 was not presented in a scientific way. I do not understand what each curves represents for. The curves color is the same and what is the difference (the reaction time) should be presented.

Figure 3 has been improved to include axes and degree of conversion data, as well as a a more descriptive figure caption (see lines 339-401).

6

According to the mdpi guide ( https://www.mdpi.com/reviewers#Review_Report), most of the cited references are better within 5 years. For the author's case, only 11 of 69 references is within 5 years. Most of them are the 1990s or 2000s papers. The authors should cite more recent papers in photo 3D printing or replace some old papers.

The authors have extensively reviewed the cited references, updating claims with more recent references where possible.  We have now included more references in the photo 3D printing space, LFAM 3D printing space, novel photoinitiator development, etc.  Of the 103 references, now 57 of the references are within the last 5 years.

7

Any reference for Figure S1?

The authors have added relevant references to support this Figure.  The Figure was produced by the authors using ChemDraw 20.0, based on information from these references.

8

The novelty of this work should be better presented. The photobleaching PIs have been reported a lot before as the authors wrote in their introduction. What is new of this BAPO PIs should be discussed.

The key contribution of this paper is not just on the utility of BAPO as a potential PI (which has been well established), but specifically on the potential utility of an optical model as a pre-screening tool in developing formulations for LFAM application based on Type I PIs such as BDMB and BAPO.  The conclusion has been modified somewhat in the first paragraph (see line 696 and 698)  to emphasize that while optical models were proposed in the 2000s, no subsequent studies are known to the authors that directly compare the optical model predictions to experimental results.  The findings of this study identify some of the shortcomings of the optical model by demonstrating that BDMB, despite non-photobleaching behavior, was able to cure through 3 mm thickness and achieve comparable mechanical properties to a BAPO formulation for the same UV dose.  Ultimately, this paper demonstrates that it is not the photobleaching behavior predicted in the optical model that most strongly influenced the rapid through-thickness cure response of a BAPO formulation, but rather the lower optical density.  The authors present a simpler pre-screening approach based on optical density, which could be further validated in future work with other Type I PIs.

9

Following question 8, the authors could make a table and discuss about the different PIs (some examples just fine) in 3D printing with their work in different parameter (bleaching or not, absorption, etc.) and polymerization and 3D printing results. This could help better solve the question 8.

The authors appreciate this suggestion to improve the manuscript and feel that this discussion of a range of PIs would be useful to explore in a future paper.  For the present manuscript, we would prefer not to include the table because we want to keep the scope focused on the optical model and the observed differences between the model predictions and experimental results.  Additional tables would also contradict the comments from another reviewer, who specifically requested a reduction in figures (other than the new Figure 3).  

Reviewer 2 Report

In this work, Stiles and colleagues studied the use of BAPO as a photoinitiator for 80:20 Bis-EMA:PETIA blend cross-linking.  BDMB was used as the control.  The authors accompanied the reactions using UV/Vis and FTIR (short and long-irradiation studies were conducted) and evaluated the mechanical properties of the composites.

In the end they show that BAPO’s performance is not inferior to BDMB.  I believe this study is interesting, well conducted and relevant for future materials science applications.  I believe the authors greatly improved the quality of the manuscript, but some critical points need to be addressed before publication:

1. Explain how the mechanical properties parameters (flexural strength, flexural modulus, failure strain) were calculated in the materials and methods section.

2. The representative representative FTIR spectra (Figure 3) is lacking axis representation and identification.  I believe this should be added to improve readability by all readers.

3. The statistical analysis presented in former phrases 444-448 should be replicated in the respective figure through the use of “*” to indicate statistical significance..

Author Response

The authors would like to thank the reviewer for their input and help in improving this manuscript.  We have added the calculations used in the flexural testing and improved Figure 3.  The third comment was unclear, and the authors would like to request further clarification in order to address this final comment.  Please see the attachment for a more detailed description.

Reviewer 2 - Where possible, all changes are highlighted in blue in the manuscript

Reviewer Comment

Authors Response

1

Explain how the mechanical properties parameters (flexural strength, flexural modulus, failure strain) were calculated in the materials and methods section.

The authors have added a paragraph to the flexural testing section (3.7) that now explains the equations and methods used to calculate the flexural properties.  Due to the brittle failure of the polymers, both flexural stress and flexural strain were reported as the maximum values from the stress-strain curves.

2

The representative FTIR spectra (Figure 3) is lacking axis representation and identification.  I believe this should be added to improve readability by all readers.

Figure 3 has been improved to include axes and degree of conversion data, as well as a more descriptive figure caption.

3

The statistical analysis presented in former phrases 444-448 should be replicated in the respective figure through the use of “*” to indicate statistical significance.

It is unclear to the authors what the reviewer is requesting.  In the original draft submitted by the authors, the lines 444-448 read:

"Ultimate DC below 100% is typical for glassy acrylate polymers, as radical occlusion can surpass bimolecular termination by a ratio of 4 to 1 during vitrification [21].  Trapped free radicals in a glassy system reportedly exceed active (e.g., mobile) free radicals at as low as 36% DC [37]. The Bis-EMA: PETIA blend in the present study is a glassy polymer system as indicated by its (meth)acrylate chemistry" 

Could the reviewer please clarify their request with a specific figure number, updated line numbers as needed and a more detailed explanation of their request?

Round 3

Reviewer 1 Report

The authors answered all my questions very well. The current version is in a good shape and is ready to publish.